# A Context-Aware Route Finding Algorithm for Self-Driving Tourists Using Ontology

**Maryam Barzegar [1,†], Abolghasem Sadeghi-Niaraki [2,3,\*,†] , Maryam Shakeri [2] and Soo-Mi Choi [3]**

[1]   Centre for Spatial Data Infrastructures and Land Administration, Department of Infrastructure Engineering, The University of Melbourne, Melbourne, VIC 3010, Australia
[2]   Geoinformation Technology Center of Excellence, Faculty of Geodesy & Geomatics Eng., K.N. Toosi University of Technology, Tehran 19697, Iran
[3]   Department of Computer Science and Engineering, Sejong University, Seoul 143-747, Korea
\*   Correspondence: a.sadeghi.ni@gmail.com
†   These authors contributed equally to this work.

**Abstract:** This study proposed a context-aware ontology-based route finding algorithm for self-driving tourists. In this algorithm, two ontologies—namely drivers' experiences and required tourist services—were used according to tourist requirements. Trips were classified into business and touristic. The algorithm was then compared with Google Maps in terms of travel time and travel length for evaluation. The results showed that the proposed algorithm performed similarly to Google Maps in some cases of business trips and better in other cases, with a maximum 10-min travel time difference. In touristic trips, the capabilities of the proposed algorithm were far better than those of Google Maps.

**Keywords:** touristic trip; business trip; ontology; experience; route finding

---

## 1. Introduction

Various tours are organized worldwide for travelling to different countries, offering a general travel plan for tourists and neglecting individuals' interests. Hence, the number of tourists interested in independent travel and personalized plans have been on the rise. In addition, most such tourists are also interested in self-driving and sightseeing in destination cities by renting cars.

Using the information on the internet, independent tourists are able to learn about the sights and destination cities and select some to visit. Independent tourists need daily planning so that they can make the most of their trips. To this end, various tourist applications have been offered for travel planning. For example, in [1], the authors offered a mobile agent, known as the Dynamic Tour Guide (DTG), which plans travel according to the user's requirements. This application mainly focuses more on navigation within tourist destinations but less on moving from one tourist destination to another. For instance, if the tourist is interested in a church, the application shows the entrance and other parts of the church, providing some information for each part through a headset. The implementation has not been provided in this paper. In [2], the authors offered key factors for designing and implementing a virtual travel operating system. The services include finding the most appropriate hotel according to the tourists' needs, selecting the best flight according to some criteria, navigation and so forth. Navigation is assumed to offer a route according to the tourists' location, needs and defined destination. However, the paper has not provided the implementation. In another research, different business programs for tourists' travel planning have been investigated [3]. The programs offer a general tour. For example, a general tour is planned for Italy according to the user's need. It offers where to start and which cities to visit. These programs have not focused on tourist navigation within a certain city.



In [4], the authors offered an ontology for saving cultural heritage and travel planning information. This model combines several ontologies and creates a general one in order to offer a smart space-based service for various modules. This is a general model, covering intercity and city to city tours; however, route finding has not been discussed. A general schematic with no implementation is offered in this paper.

To get a general view of tourist recommender systems, in [5], the authors examined various smart recommender systems for tourists, most of which take advantage of ontology; however, planning, not route finding, was the center of these systems. First, the interface used by these systems is evaluated, which is mainly web-based. The system is useful for planning prior to travel and has not focused on mobile-based systems for daily planning in destinations. Then the system is evaluated in terms of performance. Most of these programs offer sights according to the user's needs. More advanced programs provide information in terms of location and the opening time of recreational and historical centers. The investigations show that context has been taken into account in recent studies and artificial intelligence (AI) and ontology are used.

In another study, a web-based system—known as SigTur/E-Destination—providing sights' information and an appropriate tour for visiting according to the user's interest and other factors, such as a travel budget, using the Google Maps API has been offered [6]. Moreover, in [7], the authors offered a route-finding recommender system for self-drive tourists. They used Vehicle-to-Vehicle Communication (V2VCs) for collecting and sharing information. Using this technology requires vehicles equipped with V2VC in tourist hotspots in order to collect real-time information. V2VC equipment can record the nearby traffic in any POIs or share with nearby vehicles. Therefore, a self-driving tourist can obtain the real-time information regarding the road (such as distance, traffic and road conditions) and POI information (such as the current queuing time of POI) before arriving at POI. The system consists of three main modules. The first module is the data collection. The second is the scoring module that scores routes using the fuzzy theory and the TOPSIS method for optimal route finding and the third is the route generation module, providing an optimal route according to the individual's needs.

Another tourist system called SAMAP has been offered by [8], which is based on an ontology. It has information about the destination, such as sights and transportation, personal information, the user's interests, sights that other tourists are interested in and basic services a tourist needs for daily activities, such as credit card payment, taxis and so forth. Travel planning is carried out by the information. Two modules are considered for transportation (public transportation and walking) and the A* algorithm is used for this purpose. In this study, the tourists are not considered as self-driving and route finding has not been investigated.

Regarding the mentioned works, most of the tourist programs have not focused on the city tour, route finding and self-driving tourists. Only one study considered tourist as self-driving, which requires V2VC that is not possible in all cities. In addition, other details, such as context, has not been taken into account. According to [9], any information that can be used for determining the status of an entity is a part of the context. The context model distinguishes among the three system aspects of the general status: Some services are temporarily unavailable, device status (battery or memory status) and the environment (rain, traffic, temporary closure of some museums, etc.) [10].

Neglecting important factors, such as traffic and the geographic coordinates of the POIs and user's interest, causes waste of time by using inappropriate routes and arriving at the destination late or the tourist cannot visit all the interesting sights. Without considering this information, the system's response may not be tailored to the tourists' needs. As a result, the tourists are psychologically annoyed, which might result in an unpleasant memory of the destination. Consequently, the tourist is not interested in repeating travel to this destination.

Ontology is used in most tourist planning systems. Therefore, ontology is known as an efficient method in this regard. Since tourists are unfamiliar with their surrounding areas, they need to find the nearest restaurant or hospital or plan to make the most of their travels in their short travel time. Therefore, the system must be capable of responding to these needs. In addition, it needs to offer

an optimal route to the facility. As a result, categorized and sorted services that tourists need is required in the first step. An ontology can be used to save and model the service centers and their information. After finding the nearest facility, it is essential to generate a route. The experience of the locals, who are experts in finding optimal and low-traffic routes in terms of time and distance, is an appropriate choice for the offline route finding without V2VC. Such experiences can be collected, modeled and used to improve route finding.

Various researchers have taken advantage of drivers' experiences in route finding. For example, in [11], the authors created a hierarchical network for route finding using GPS data from taxis and they only considered the data from peak traffic times. In [12], GPS data from taxi drivers has been used in order to create an experimental network. In this study, the taxi speed must not be less than a minimum in order to separate the unnecessary GPS data such as slow driving taxis looking for passengers. In addition, a hierarchical method has been used for route finding. First, the existence of the location and destination in the experimental network is checked. If both are available in the experimental network, this network is used for route finding. If either does not exist, the nearest point(s) is (are) found in the network and a rectangle with a diagonal longer than the distance between the point (location or destination) and nearest point (to location or destination) in the experimental network is selected. For example, if both location and destination do not exist in the network, route finding is carried out as follows, using the Dijkstra's algorithm: The route between the location and the nearest point to the location in the experimental network, the route between the nearest point to the location in the experimental network and the nearest point to destination in the experimental network, the route between the nearest point to destination in the experimental network and the destination. In the current paper, a part of this hierarchical method has been used.

In another study, the authors used GPS data from taxi drivers to create a time-dependent graph of experimental routes. In this method, a node, which is a road segment frequently used by taxi drivers, is used. They also used an entropy-variance clustering method to estimate the duration [13]. In [14], the authors used taxi GPS data to create a graph. They used this graph to find the shortest route with the minimum cost. According to the investigations, the experiences of taxi drivers are found to be useful for improving route finding. In addition, there have been other studies that used heuristic optimization algorithms, such as Genetics and PSO; some of them have been reviewed in [15]. These methods that can find a feasible path with a very high probability [16] have not been discussed in this study. This study uses the optimal route algorithm for example, Dijkstra that exactly minimizes the sum of the edge weights to find an optimal path [16,17] as a part of the proposed algorithm.

Another important point is modeling and formulating experiences. Experience is knowledge and there are various methods to formulate the knowledge and ontology is one of them. Many researchers have used the ontology to formulate knowledge and store experiences. For example, in [18], the authors examined the role of organized information in the process of organized learning. The knowledge in this paper is presented by using ontology. Ontology is the basis for the knowledge map and web-based system performance assigned to organized learning. The innovation of the web system in the current paper is organizing resources by the knowledge map. The project, defined in this study has focused on collecting knowledge and skills in terms of organizing and more accurately, collecting the knowledge-related resources. The purpose of this project is the management of these resources through an IT operating system, which supports the organizational memory of its users. The study also focuses on how the users of an organization can utilize the system as the organizational learning vector.

In another studies such as [19–21], the authors have also used ontology, knowledge base and web software for exchanging experiences. Moreover, a knowledge base for collecting experiences and graphs for displaying ontology in industry have been used in [22]. Ontology is found to be an efficient method for formulating knowledge. Experiences can be explicitly formulated and shared. Whenever needed, the system can be updated, and new experiences are added.

This paper aims to propose a context-aware route finding algorithm based on ontology models of driver's experiences for self-driving tourists. This algorithm considers two kind of trips including

business and touristic trips. In the proposed algorithm, if the location and destination are in the ontology of drivers' experiences, the optimal route between location and destination will be presented for the user by retrieving route segments from ontology without any processing and if the location or destination does not exist in the drivers' experiences ontology, a part of route will be calculated based on ontology and another part based on Dijkstra algorithm. In this regards, two ontologies are designed in this study (the ontology of drivers' experiences and the ontology of required services for tourists). After designing both ontologies, the data are collected for each of them. The data needed for the ontology of drivers' experiences are collected by an application. The data needed for the services ontology are collected by OSM and located in the related class. After collecting data, a route finding algorithm is designed according to both ontologies. Finally, the algorithm is implemented for Tehran, Iran and compared in two modes (peak and off-peak) using Google Maps in terms of travel time and travel length.

The organization of this paper is as follows: Section 2 of the paper is the proposed method for route finding, covering the explanations for the drivers' experience ontology, services ontology and context-related information. Finally, the context-aware route finding algorithm is proposed according to the drivers' experience ontology and services ontology. In Section 3, the proposed method is implemented for business and touristic trips. Finally, the results are compared with Google Maps in terms of travel time and travel length Section 4. Section 5 provides the conclusion.

## 2. Method

To propose a route finding algorithm for self-driving tourists, the ontology model is used to find a route in real-time based on user needs in every condition. As route finding algorithms usually take long time to find optimal route between two points due to the large amount of data (Shen and Ban, 2016), modeling driver' spatial experiences using ontology facilitates the timeliness of services and provides the capability to make use of earlier information without re-processing. Moreover, the advantage of using ontology for storing data in OWL files is to enhance the sharing, reusing and processing of domain knowledge [23,24].

Two ontologies are used for urban route finding. Drivers' experience ontology models the experiences of drivers and the service ontology presents the services tourists need through an ontology. In general, two routes may be requested by tourists: a business trip which is from a source and destination and a touristic trip to visit tourist destinations on the way. If a business trip is requested by the tourist, the driver's experience ontology and route finding algorithm are only used but in the case of a touristic trip, both ontologies are used.

### 2.1. Drivers' Experiences Ontology

Figure 1 shows the drivers' experiences ontology. This ontology has two main classes of experimental and non-experimental routes. Each of these classes covers two subclasses of traffic and non-traffic. The experimental route class and its related subclasses are routes that drivers have traveled at peak traffic times (7–10 a.m. and 4–8 p.m.) and off-peak traffic times. Each drivers' route is defined by a separate class. According to the departure time, it is also defined as a subclass of traffic or without traffic experimental routes classes. If the route is a subclass of the 'without traffic' class, the name of the class will be location_destination and if it is a subclass of the 'traffic' class, the class name will be location_destination_t, in which t represents the traffic. The individuals of this class are route segments and the ID of each route segment is the individual's name. X and Y coordinates are the individual's data properties. For example, if the driver' route between Poonak and Tajrish is considered at the traffic time, the class and OWL file named Poonak_Tajrish_t is created as a subclass of traffic class in drivers' experiences ontology. In this class, each segment of the route, which should be traveled to reach Tajrish from Poonak at the peak traffic time, is defined as an individual. The individual of the first segment of Poonak_Tajrish_t class is as follows:

<owl:NamedIndividual rdf:about="http://www.co-ode.org/ontologies/Poonak_Tajrish_t.owl#0">
      <y_coor rdf:datatype="http://www.w3.org/2001/XMLSchema#integer">3960467</y_coor>
        <x_coor rdf:datatype="http://www.w3.org/2001/XMLSchema#integer">535432</x_coor>
</owl:NamedIndividual>

Non-experimental routes include routes obtained through the route finding algorithm. According to the departure time, each new route, generated by the route finding algorithm, is defined as with traffic or without traffic subclasses of the non-experimental routes class. If the route is a subclass of the 'without traffic' class, the name of the class will be location_destination_n in which n represents non-experimental and if it is a subclass of the 'traffic' class, the class name will be location_destination_n_t. The individuals of each route are defined in a separate file as a class with a similar name to its route class in the drivers' experience ontology. The reason for this work is preventing the ontology from complexity and increasing the search speed in the ontology file.

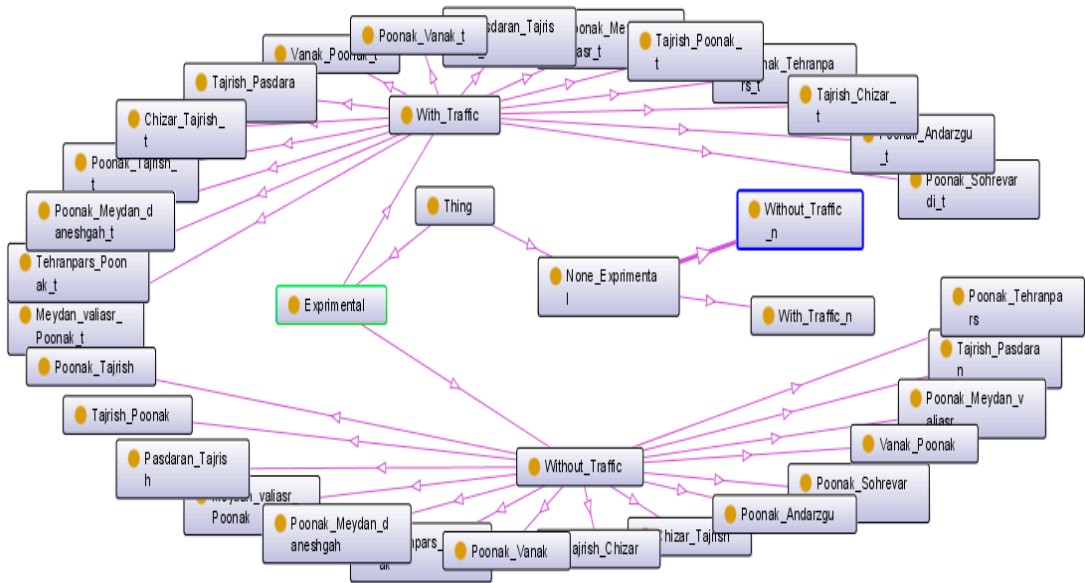

**Figure 1.** Driver's experience ontology.

## 2.2. Ontology of Required Tourist Services

This ontology, covering various classes that tourists may need during the travel including shopping (bakery, commercial, mall, supermarket, etc.) and food (bar, café, fast food, food court, restaurant, etc.). Table 1 lists the classes and subclasses of this ontology and Figure 2 shows ontology of required tourist services.

**Table 1.** Service ontology classes and subclasses.

| Class Name | Subclasses |
|---|---|
| Accommodating | Hostel, Hotel, Motel |
| Attractive places | History (Museum, Memorial, Monument, Manor, Tower (Like Azadi in IRAN), Bridge), Zoo, Viewpoint (Like Milad Tower in Iran), Skyscraper |
| Car related activities | Car Rental, Car Wash, Fuel, Parking |
| Disaster | Camp Site, Fire Station, Police, Shelter |
| Education | College, Dormitory, Institute, Kindergarten, Language School, Library, School, University |
| Food | Bar, Café, Fast Food, Food Court, Restaurant |
| Health | Dentist, Hospital, Pharmacy, Veterinary |
| Information services | Currency Exchange, Embassy, Information |

**Table 1.** *Cont.*

| Class Name | Subclasses |
|---|---|
| Leisure | Forest, Park, Picnic Site, Toilet |
| Public places | Administrative (Bank, Community Center, Court House, Post Office, Prison), Art (Art Center, Cinema, Concert Hall, Gallery, Studio, Theater, Town Hall), Entertainment, Worship (Mosque, Shrine, Church, Temple) |
| Public transportation | Bus, Subway, Taxi |
| Shop | Bakery, Commercial, Jewelry, Mall, Super Market |
| Sport | Golf Course, Ice Rink, Pitch, Stadium, Swimming Pool, Tennis Court, Sport Center |

Since tourists might not be familiar with the environment, they might face some accidents, not only with the services that tourists need during normal trips, such as food and accommodation but also some services needed after accidents such as vehicle breakdown, fire, theft and so forth, are considered. Other facilities, such as sports, generally provided in gyms (bodybuilding, aerobics, etc.) are all classified as the sport center class. Other sports which need fields such as golf are separately classified.

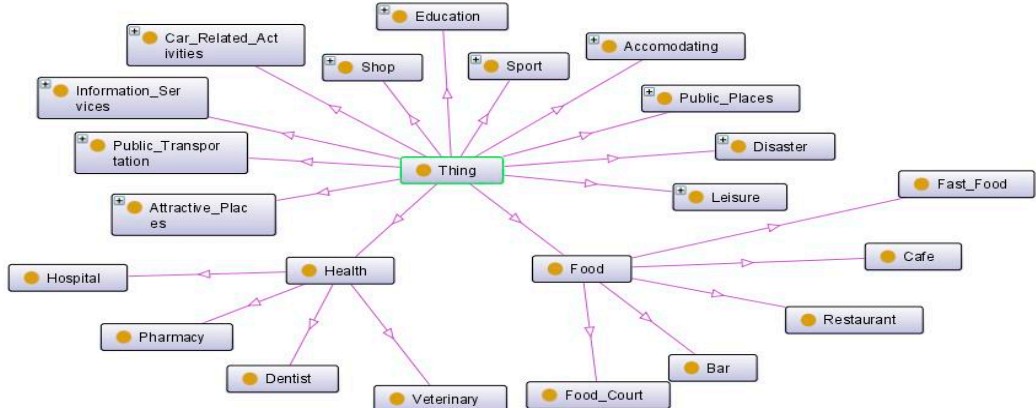

**Figure 2.** Ontology of required tourist services.

## 2.3. Context-Related Information

Either the user or the data-centric approach can be considered in every system. In the data-centric approach, GIS is designed according to the data after collecting service data by the experts and the results are shown to users as outputs. Therefore, since the user's preference is not considered in the system design, the output might not be optimal from the user's perspective. However, in the user-centric approach, the user's preference is considered in the system design, and services are offered according to the tourist's interests [25]. Here, the results would vary for users with different preferences.

In this regard, user-related information can be considered part of the context. According to [26], context-related information is assigned into personal and environmental context. Figure 3 shows the context information. The personal context includes information such as user's preference and user's position. Preference information is considered as history or interest. If the tourist is interested in history, history-related classes can be searched in the Attractive Places class. On the contrary, if the tourist is not interested in historical places, other subclasses, not related to Attractive Places, are searched. In the Interest Section, tourists can enter their interests in terms of accommodation, food and sport. Therefore, while requesting food, the tourist's preference is taken into account and appropriate food-related places are offered. Such information acts as filtering. In addition to increasing the speed, system outputs are ensured to be compatible with the user's need and interests. In the user's position section, the tourist's location information is saved using GPS. In addition, the user's position can be considered as the starting point of the trip.

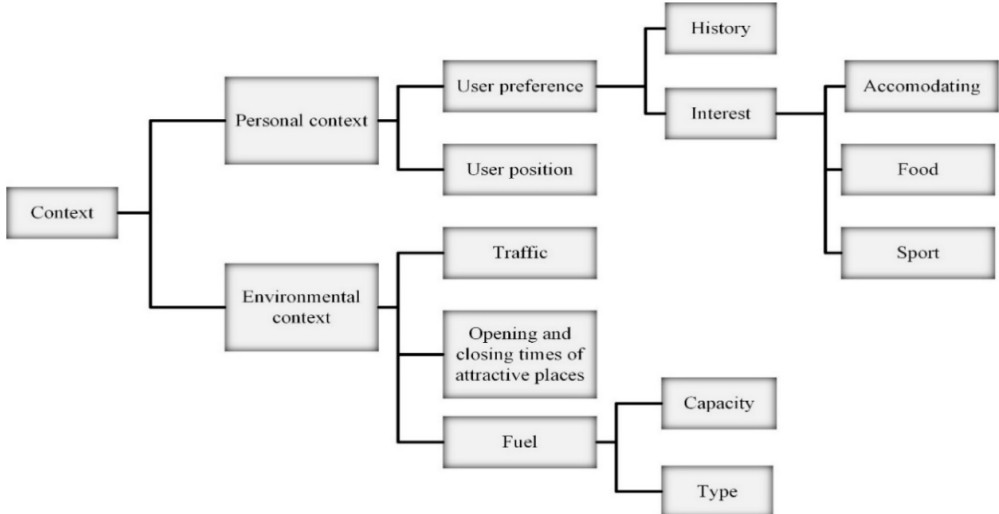

**Figure 3.** Context-related information.

In the environmental section, information related to traffic, opening and closing time of museums and tourist places, fuel capacity, type of fuel and departure time are considered. Since the routes of experienced drivers are used and the data are collected at peak and off-peak traffic times, the routes contain traffic information. The other information is used while searching for the user's requirements in the service ontology. For example, when the car's fuel is low and the required fuel is gas, the system only considers fuel centers that have gas fuel. In case of requesting recreational centers, only open centers are taken into account according to departure time.

*2.4. Context-Aware Route Finding Algorithm Based on the Driver's Experiences and Service Ontologies*

In this paper, a network is first created and then route finding is carried out accordingly. In each graph, each route segment is considered as an edge. The edge's weight is determined based on the idea of using the frequency of segments that are traveled by drivers. This idea was proposed by [13]. Since the frequency of passing an edge might be large, the frequency of passing from an edge is normally determined using the following formula:

$$f_n(e_i) = \frac{f(e_i) - f_{min} + 1}{f_{max} - f_{min} + 1} \tag{1}$$

where $f_n(e_i)$ is the number of normal passages from $e_i$, $f(e_i)$ is the number of passages from $e_i$, and $f_{min}$ and $f_{max}$ are the minimum and maximum number of passages from the edges. It should be noted that this formula is obtained based on the Min-Max normalization method in which 1 value is added to both numerator and denominator to avoid negative results. For a segment which is not in the experimental way, $f(e_i)$ is zero. Therefore, $f_{min}$ is zero. In general, the weight of a network graph is determined as follows to reverse the values:

$$w(e_i) = \max\left(\frac{f_n(e)}{Length\ (e)}\right) - \frac{f_n(e_i)}{Length\ (e_i)} \tag{2}$$

where $w(e_i)$ is the weight of the $i^{th}$ edge of the network and the length $(e_i)$ is the $i^{th}$ edge length. $max(f_n(e)/Length\ (e))$ is the maximum number of passages from an edge divided by the edge length. Since each edge with less weight is preferred in Dijkstra's algorithm, certain values are deducted from the maximum value in order to reverse the values. Experimental Network 1 includes the routes traveled by taxi drivers at off-peak traffic times. Experimental Network 2 includes the routes traveled by taxi drivers at peak traffic times. The main network is the major layer of the urban network. The general

experimental network is included in the experimental network 1 and 2. In general, two cases are considered for the tourist route finding as follows. Figure 4 shows these two cases.

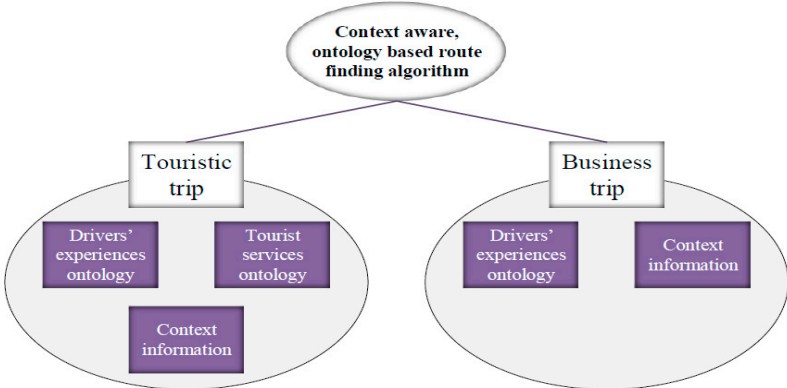

**Figure 4.** Two cases of the proposed route finding algorithm.

### 2.4.1. Business Trip

If the tourist only needs the route between the location and destination, the trip is considered a business trip. As shown in Figure 5, in this case, the drivers' experiences ontology is used. After receiving the context, location and destination information, it is essential to examine whether there is a subclass with this location and destination in the ontology or not. If so, the saved route is displayed to the user without using route finding. Otherwise, it is necessary to examine whether or not the location and destination exist in the experimental network. According to departure time, one of the Experimental Networks (1 or 2) is considered. If both location and destination are located in the Experimental Network, the experimental network is used for creating the graph and route finding. If either location or destination does not exist in the Experimental Network, the nearest point to one or both is found in the network.

```
Business Trip Algorithm
Input:   LocationName: name of location,
         DestinationName: name of destination,
         T= departure time
Output:  x coordinates of path segments,
         y coordinates of path segments,
         IDs of path segments
Procedure:

If 7<T< 10 or 16<T<20
        return Class_name = LocationName _ DestinationName_t,
    else
        return Class_name = LocationName _ DestinationName
end

If Class_name was in Drivers' Experiences Ontology
        return individuals of class_name class
        break
        If Class_name = LocationName _ DestinationName_t,
            Consider experimental road network based on traffic classes of ontology
        else
            Consider experimental road network based on without traffic classes of ontology
        end

        If LocationName & DestinationName are in experimental road network
            Consider only experimental road network for route finding
        elseIf LocationName is not in experimental road network
            Find the nearest point to location (L1) in experimental road network and extract a rectangle from main road network
        elseIf DestinationName is not in experimental road network
            Find the nearest point to Destination (D1) in experimental road network and extract a rectangle from main road network
        end
        If L1= null and D1= null
            Compute the shortest path (Location-Destination)
        elseIf L1!= null and D1 = null
            Compute the shortest path (Location-L1, L1-Destination)
        elseIf L1= null and D1!= null
            Compute the shortest path (Location-D1, D1-Destination)
        else
            Compute the shortest path (Location-L1, L1-D1, D1-Destination)
        end
        Store Class_name as a subclass of None_Exprimental class,
        Segment IDs as individual names
        x coordinates & y coordinates of segments as data property of individuals
end
```

**Figure 5.** Business trip algorithm.

As shown in Figure 6, the distance between the source (S) and destination (D) is first considered as diagonal ($d_1$) of a rectangle known as $R_1$. Then the value of h is added to $d_1$ to each side of the rectangle in order to create a large rectangle of $R_2$. Actually, the h value is used to define searching rectangle dimensions (R2) to find the nearest point(s) to source or destination in the Experimental network. If the h value is large, the processing volume is increased due to the increasing number of segments and if the value of h is small, the nearest point(s) to the source or destination may not be found. In this study, the value of h was considered 0.01 d1 which was determined based on trial and error. Therefore, the diagonal (h) of the final rectangle is d2 = d1 + 2$\sqrt{2}$ h. Finally, the larger rectangle is searched to find the nearest point(s) to the source or destination.

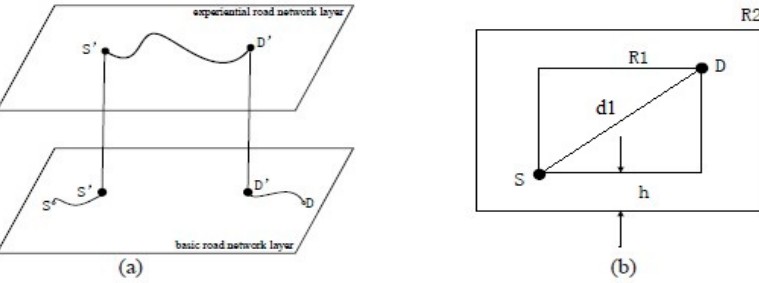

**Figure 6.** (**a**) Hierarchical Experimental Network (**b**) Definition of the rectangle for the main network Ji-Hua et al. (2013).

For example, if both source (S) and destination (D) are not within the Experimental Network (Figure 6), the distance between S and D is considered as $d_1$, which is the $R_1$ diagonal. Then the larger rectangle of $R_2$ with the diagonal of $d_2$ is formed, which is created in addition to h on each side of $R_1$. Within the larger rectangle, the nearest points to S and D are found in the Experimental Network. After finding these points, the distances between SS' using the main network, S'D' using the experimental network and D'D using the main network are calculated. These three routes are connected, and an overall route is shown to the user. After route finding, the name of the route is saved as one of the with or without traffic subclasses of the non-experimental class in the driver's experiences ontology, according to departure time, and if the user requests this route in the future, the saved route is immediately shown to the user without route finding.

### 2.4.2. Touristic Trip

If the tourist needs to stop on the way in order to use certain facilities, the trip is considered a touristic trip. In this case, both ontologies are used for route finding. For example, if the tourist needs food and then a gas station before arriving at the destination, three food centers which are near to the source are found using food-related subclasses, context data and Euclidean distance. Then the network distance to the source is calculated for these three centers and then the nearest center to the source according to the network distance is determined as the first stop. Therefore, the algorithm calculates a route between the source and this center using the ontology-based route finding algorithm.

In the touristic trip, the ontology-based route finding problem is based on the concept of divide and conquer approach [27]. With this approach, the problem of route finding is divided into several small scale sub-problems and then each sub-problem is solved, respectively. After that, the results of sub-problems connect together to create the overall route. The first stop is considered as the new source and the gas station as the second stop. The Euclidian distance is used to find the three gas stations near the first stop. Then the network distance to the first stop is calculated and the gas station with the shortest network distance to the first stop is selected as the second stop. The distance between the first and second stop is calculated using the algorithm. Finally, the route between the gas station (second stop) and the destination is calculated. All three routes (source-first stop, first stop-second stop and second stop-destination) are connected and shown as a route to the user.

Note that each of the three routes is considered as a separate business trip. The number of services can vary and is not limited to two, according to the abovementioned example. The touristic trip is generally time-taking and the tourist might stay hours in each place. Therefore, if the tourist requests route finding in peak traffic times and then enters the off-peak traffic times in the middle of the trip, the system can warn the tourist that the time duration for using this route is over and re-routing is required. In the next section, the required data and applications for the implementation of the proposed method will be described. Figure 7 shows the touristic trip algorithm.

## Touristic Trip Algorithm

```
Input:    LocationName: name of location,
          DestinationName: name of destination,
          S: vector of required services during the trip
          T= departure time
Output:   x coordinates of path segments,
          y coordinates of path segments,
          IDs of path segments
Procedure:

Location = LocationName
For all s[i= 1:n] ∈ S
        Find the related subclass in Tourist Services Ontology
        Find three nearest individuals of the related subclass based on Elucidation distance
        Calculate the network distance from Location to these three individuals
        N= The nearest individual to Location based on network distance
        Destination = N
        R[i]= Find the shortest path from Location to N with Business Trip Algorithm
        Location = N
end
```

**Figure 7.** Touristic trip algorithm.

## 3. Implementation

The shapefile layer of Tehran Open Street Map (OSM) is used as the main network. To create the experimental network, the data are collected using an application with the OSM layer of the city of Tehran for saving the routes traveled by drivers. This application is programmed in an Android environment using ArcGIS SDK for Android Spatialite. The information is saved in the form of GPS points in a database and the location and destination are saved as text format. The routes traveled by over 30 experienced taxi drivers of Tehran at off-peak and peak traffic times in different routes (different sources and destinations) are collected by the application and converted into a shapefile. After pre-processing, Experimental Network 1 and 2 are created in ArcGIS using these routes. Then Experimental Network 1 and 2 and the main network are saved in an Oracle database.

To implement the drivers' experiences ontology, the overall ontology is designed in Protégé. Then each route is separately saved as a class in a separate file with a similar name to the related class in the drivers' experiences ontology using OWL API java. If a new route is created in each run of the algorithm, which is not available among the routes generated by the ontology, it is saved as a class in order to show to the user without processing if requested in the future. To retrieve the routes from the ontology, pellet reasoner is used as a reasoner and OWL reasoner as an interface for OWL API. This program, which is written in desktop using java, converts each new route into a separate OWL class. Therefore, whenever the program is run, the ontology is updated, and new routes are added. As a result, data processing declines over time.

The services ontology is also designed in Protégé. The data related to each class is collected by OSM data in Tehran and then saved in the Oracle database. The data properties of each facility, which are X and Y coordinates, are used to find the nearest facility to the previous source. The name of each individual in the service ontology subclasses is determined using its original name in OSM. If the user requests a facility such as a park, the nearest park is first found. Then the route is calculated to

reach this park. Another route is then found from the park to the destination. After connecting these two routes, the overall route is shown to the user.

## 4. Evaluation

There are various methods to evaluate an ontology. For example, according to the Gomez-Perez approach [28,29], an ontology needs to meet certain criteria, structural considerations need to be taken into account for organizing an ontology and the appropriateness for reusing and redeveloping should be investigated. To evaluate the ontology reuse, ontology must be capable of being used for another application. According to this approach, the drivers' experiences ontology must be capable of being used in other countries. For designing experiences ontology, the coordinates of the start of the ID for each segment are required to distinguish other segments. These three features are available in all databases of the road network. Even if the data is not OSM data, and, for example, the saved GPS points in a database being used, these features are also accessible in GPS points. Therefore, the database of drivers' experiences ontology database can be easily altered and used for another purpose. Service ontology is also general and can be used anywhere. Broadly speaking, depending on the ontology type, there are four types of evaluation [30]: evaluating the results of an application in which ontology is used [31], comparing the ontology with a database according to the domain which that ontology is covering [32], evaluation of ontology by experts to see whether or not the ontology under evaluation meets certain pre-defined criteria [33] and comparing the ontology with a standard golden ontology [34].

Among these methods, since an ideal database and standard golden ontology is not accessible, the first and third can be used to evaluate the drivers' experiences ontology. The third method is mainly used to compare two or more ontologies. It evaluates the criteria according to the use of tools in ontology (Edit, compatibility check, use of other languages, etc.), language (the possibility of defining features for each class, the possibility of defining cardinality for features, the possibility of defining functions, etc.), content (Existing concepts, number of relationships, etc.), method (Clarity of activity and method description, the number of fields, etc.) and cost (the cost of access to interface, permissions for ontology tools, etc.). These criteria are defined by experts and weighted using AHP and then the results are compared [35,36]. Therefore, the best way for evaluating the drivers' experiences ontology is the first method. The application-based evaluation examines how much results of the application are influenced by the ontology. The results might get worse or better by an ontology [37]. To evaluate the drivers' experiences and services ontologies using the first method, the route finding application is taken into account and the route finding algorithm is compared with Google Maps. As the Google Maps navigation system uses traffic information, the routes which were calculated by Google Maps with traffic information are compared with the routes of the proposed algorithm in both traffic times and non-traffic times. In this regard, the fastest route of Google Maps is used to compare with the results of the algorithm based on time and distance. As the comparisons between the proposed algorithm and Google Maps are done in the same departure times, they have same traffic conditions. The results are provided in the next section.

### 4.1. Application-Based Ontology Evaluation

To evaluate the proposed model, it is examined whether only routes that are traveled at peak traffic times can improve the route finding results at peak traffic times or not. In this regard, ten pairs of sources and destinations, which almost cover the whole of the city of Tehran, are selected. First, the route finding algorithm is used according to the Experimental Road network (ER). Then the route finding algorithm is run according to Experimental Network 1 and 2 (ETR). Departure time is also selected at peak traffic times. For all routes, the travel length is calculated by the total sum of the segments' lengths. Travel time is also calculated by ten drivers traveling routes. Figure 8 shows the travel length for ten routes. Figure 9 shows the travel time of 10 pairs of location and destination for two cases of the proposed method.

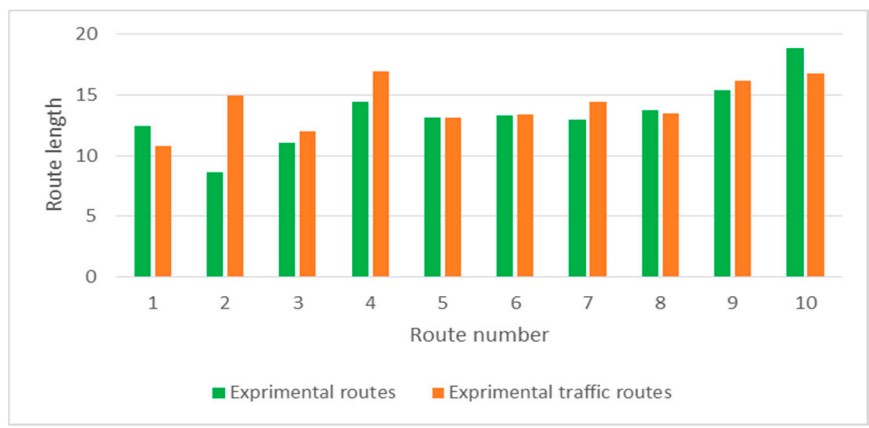

**Figure 8.** Comparison of travel length for two cases of the proposed method.

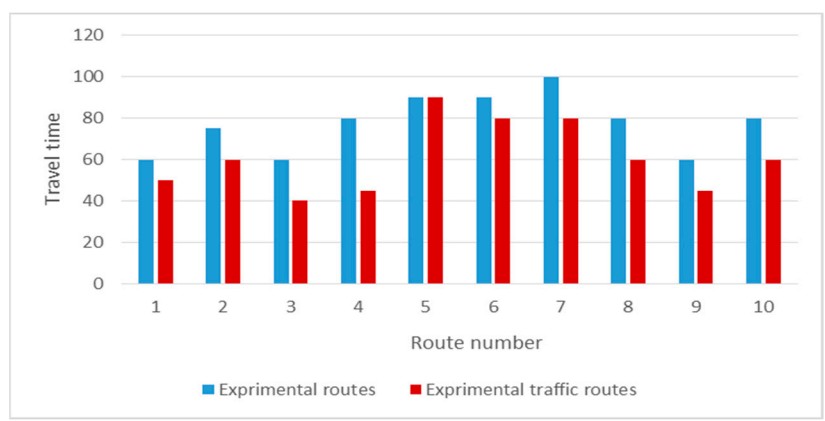

**Figure 9.** Comparison of travel time for two cases of the proposed method.

According to Figure 8, the travel length in ETR mode is longer than in five cases, equal in two cases and shorter in three cases. According to Figure 9, the travel time in ETR is shorter than ER in 9 cases and equal in one case. In general, although the travel length is greater in ETR than ER in few cases, the travel time is shorter. Therefore, considering only routes that are traveled at peak traffic times significantly helps the optimal route finding and results in shorter travel time. In the following section, the calculations are made according to the Experimental Network 1 for peak traffic times and Experimental Network 2 for off-peak traffic times.

### 4.1.1. Evaluation of the Proposed Method for Business Trips

Here, the tourist is assumed to have a trip between the location and destination with no stops. For evaluating this case, nine pairs of other sources and destinations at peak, and five pairs in off-peak traffic times, are considered. The routes are calculated using the drivers' experiences ontology-based algorithm and Google Maps. The results are then compared in terms of travel length and travel time. Since route finding is of great importance in peak traffic times, more routes are considered at peak traffic times than the off-peak traffic times. Figure 10 shows the resulting route of the drivers' experiences ontology-based algorithm compared to Google Maps for Route 4, related to location and destination pairs traveled at off-peak traffic times. Figure 11 compares the result of the proposed algorithm and Google Maps for Route 2, related to the location and destination pairs traveled at peak traffic times.

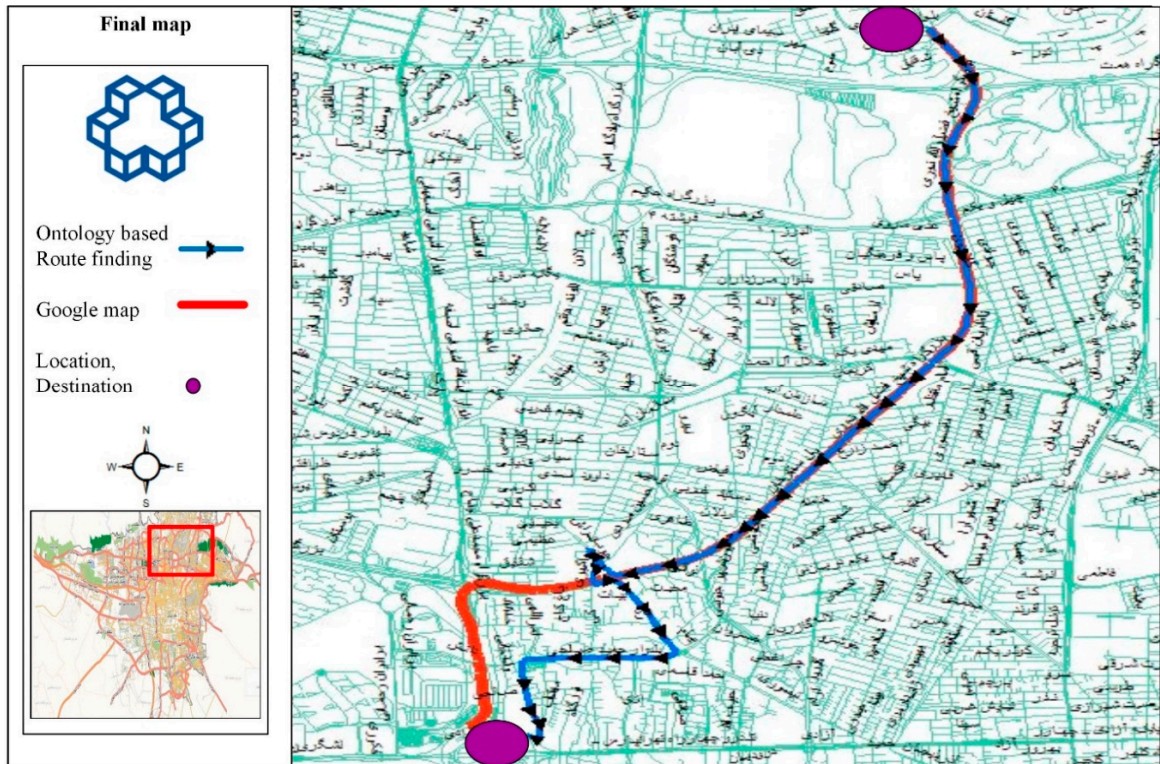

**Figure 10.** Comparison of two methods at off-peak traffic times.

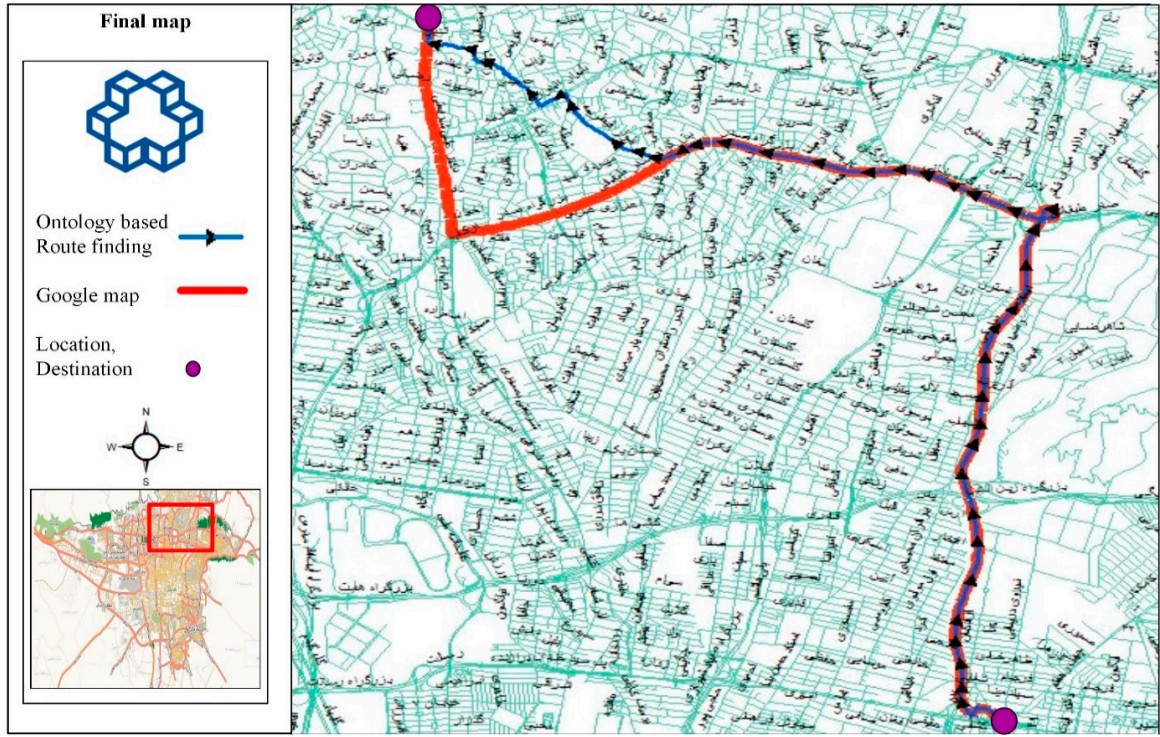

**Figure 11.** Comparison of two methods at peak traffic times.

Figures 12 and 13 compare the travel length and travel time of the proposed algorithm and Google Maps for five pairs of location and destination at off-peak traffic times. According to Figure 12, the route length is found to be equal in ETR and Google Maps in two cases and shorter in one case. According to Figure 12, the ETR's travel times are shorter than Google Maps in three cases and equal

in two cases. Note that the resulting route of the proposed algorithm is completely similar to Google Maps in routes where travel times and travel lengths are similar (Route 1 and 5).

According to Figure 14, the ETR's travel length is shorter than Google Maps' in three cases, equal in three cases and, longer in three cases. According to Figure 15, the travel times of ETR are equal in four cases and shorter than Google Maps in five cases. Note that Routes 4, 7 and 9 are similar in both modes. According to the results, the proposed algorithm functionality is found to be better than Google Maps in the majority of routes and, the travel time difference reaches up to 10 min in some cases. These differences are due to that experienced drivers using some secondary roads. However, using these roads might lead to increasing the travel length but reducing the travel time due to less traffic. Therefore, using drivers' experiences and creating a road network with routes that are traveled by experienced drivers can improve the route finding performance.

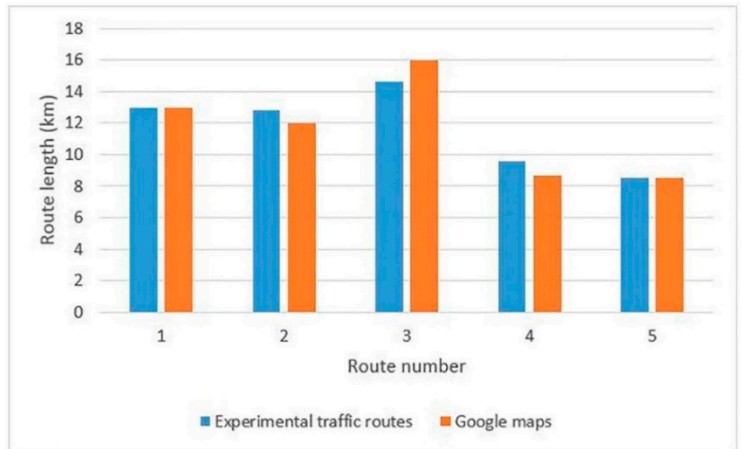

**Figure 12.** Comparison of route length for the proposed algorithm and Google Maps at off-peak traffic times.

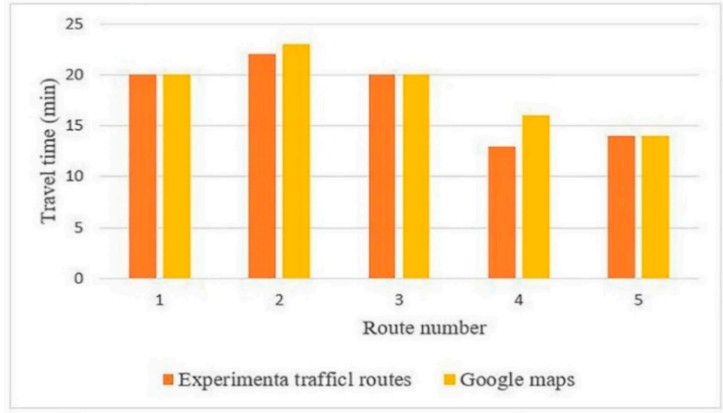

**Figure 13.** Comparison of travel time for the proposed algorithm and Google Maps at off-peak traffic times.

Figures 14 and 15 compare the results of the proposed algorithm and Google Maps for ten pairs of location and destination at peak traffic times in terms of travel length and travel times.

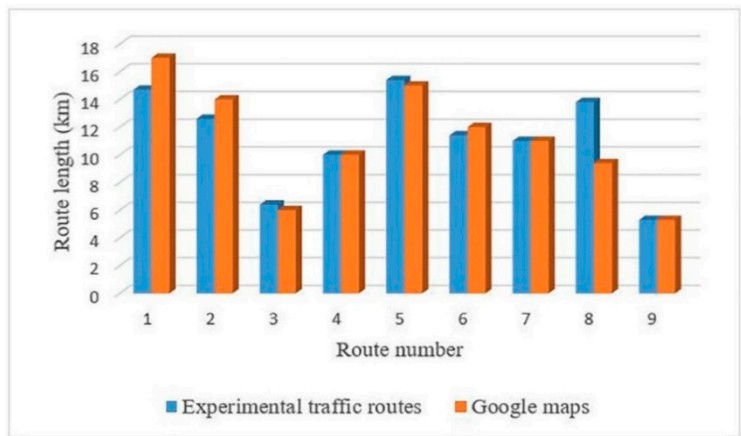

**Figure 14.** Comparison of travel length for the proposed algorithm and Google Maps at peak traffic times.

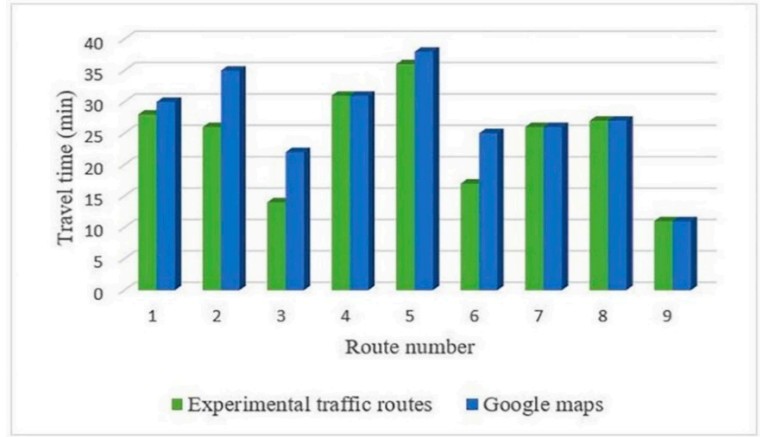

**Figure 15.** Comparison of travel time for the proposed algorithm and Google Maps at peak traffic times.

4.1.2. Evaluation of the Proposed Method for Touristic Trip

If the tourist needs services and facilities on the way, after entering the location, destination and services, the route is calculated by an algorithm. To calculate the route, if the tourist wants to travel from, for example, Pounak to Tajrish and needs to go to a gas station and find food on the way, the algorithm finds the nearest gas station to the source and then the nearest food center to the gas station using the Euclidean distance and network distance and considers the fuel sub class in the car-related activities class of facility ontology and data properties of individuals. Finally, the routes from the source to the gas station, gas station to the food center and food center to the destination are calculated using the algorithm and shown to the user. Figure 16 shows the proposed algorithm performance with consideration of two stops on the way.

The difference with Google Maps lies in the fact that tourists find all facilities, such as food nearby, with Google Maps and then select which to try. Afterwards, a route is proposed from the source to the center by Google Maps. Here, tourists have to estimate which center to choose. Meanwhile, tourists are not able to choose different facilities at the same time to have a general route so that they cannot use a set of facilities for better planning. In the proposed algorithm and service ontology, all services are more accurately provided. For example, when food is needed, tourists can choose among different classifications or a center can be chosen using context information. In addition, tourists should not investigate food centers near the source and the algorithm automatically provides them the nearest centers according to context information. Furthermore, tourists are able to see and select different facilities simultaneously. Since the algorithm considers each segment (source-facility, facility-facility or facility-destination) as a separate pair of location and destination and are like a separate business trip,

due to better results that are proposed in the case of a business trip which shows that the algorithm acts better than the Google Maps in the majority of cases and are similar in other cases, it can be ensured that the resulting routes of the algorithm in the case of touristic trips are also, optimal routes.

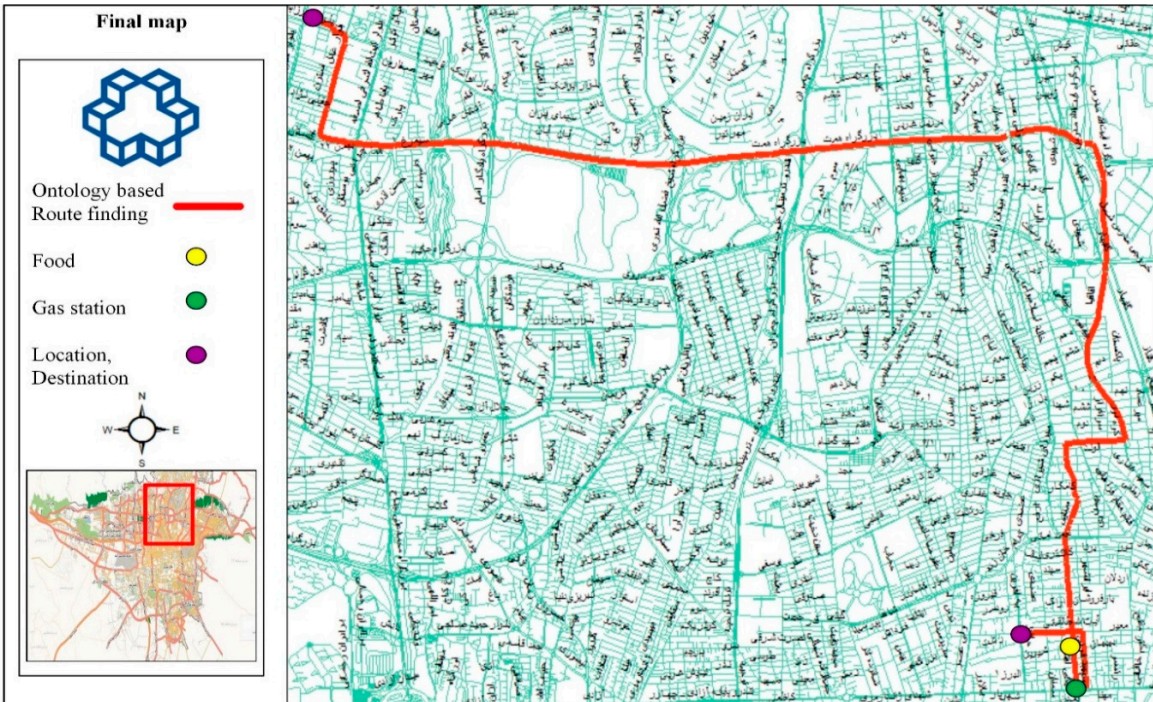

**Figure 16.** The proposed route with two stops on the way.

## 5. Conclusions

Many tourists are interested in self-driving worldwide and need to plan prior to their trips. Therefore, finding an appropriate route is of special importance due to time limitations. The purpose of this study is route finding for self-driving tourists according to their requirements and due to this object, a context-aware, ontology-based route finding algorithm is proposed. In this algorithm, drivers' experiences in route finding are saved in an ontology. Another ontology is proposed for required tourist services that enable tourists to plan for their trips. In this algorithm, a tourist can have two options: a business trip (trip between a source and a destination) and a touristic trip (with stop points between the source and destination). Therefore, the algorithm provides an appropriate route using the context information and user's needs. The difference between this algorithm and its previous counterparts is the individual modeling of drivers' experiences using the ontology and taking advantage of this ontology in route finding. With this ontology, drivers' experiences can be saved and used in the future. Also, the ontology is updated over time; as a result, there is a need to process declines. According to the evaluations, considering only traffic-related routes at peak traffic times significantly helps the improvement of route finding and reduces travel time. Further studies are required to add other capabilities to the touristic trip case. For example, other information, such as the history of touristic places and some related images, can also be shown to the users. The business trip, offered in this study, can also be used for locals. In addition, it is best to compare the results of the proposed algorithm with the routes which are obtained based on different criteria, such as with traffic or without traffic.

**Author Contributions:** These authors contributed equally to this work: M. Barzegar and A. Sadeghi-Niaraki (co-first authors). Conceptualization, M.B. and A.S.-N.; methodology, M.B., A.S.-N. and M.S.; software, M.B. and M.S.; validation, A.S.-N.; formal analysis, M.B.; investigation, M.B., A.S.-N. and M.S.; resources, A.S.-N.; data curation, M.B. and M.S.; writing–original draft preparation, M.B.; writing–review and editing, A.S.-N. and S.-M.C.; visualization, A.S.-N.; supervision, A.S.-N.; project administration, S.-M.C.; funding acquisition, S.-M.C.

**Funding:** This research was supported by the MSIT (Ministry of Science and ICT), Korea under the ITRC (Information Technology Research Center) support program (IITP-2019-2016-0-00312), supervised by the IITP (Institute for Information & communications Technology Planning & Evaluation).

**Conflicts of Interest:** The authors declare no conflict of interest.

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
