# Peer review of "A Context-Aware Route Finding Algorithm for Self-Driving Tourists Using Ontology"

_electronics, doi:10.3390/electronics8070808_

Round 1

Reviewer 1 Report

The results are clearly presented.

Author Response

Dear Prof. Dr. Mostafa Bassiouni,

Editor-in-Chief,

Electronics ,

May, 2019

Re: (Manuscript ID: electronics-512692) A Context-Aware Ontology-Based Route Finding Algorithm for Self-Driving Tourists

We thank both reviewers for providing us highly constructive and insightful comments to improve our manuscript. We have carefully revised the manuscript, following the reviewers’ suggestions, and have responded in detail to each comment. The next section contains our point-by-point responses (in blue) and changes and referencing to the manuscript (in green), based on the reviewers’ comments (in italic). We believe that our manuscript has substantially improved and is more readable for broader audiences. We would like to thank you, Prof. Dr. Mostafa Bassiouni (Editor-in-Chief), related Section Editor-in-Chief and Ms. Coleen.Shi, M.S. (Assistant Editor) for following up this manuscript. We look forward to hearing from you. Also, we would be glad to respond to any further questions and comments that you may have.

Best Regards,

Abolghasem Sadeghi-Niaraki,

################################################

Dr. Eng. Abolghasem Sadeghi-Niaraki

Assistant Professor

Faculty of Geodesy and Geomatics Engineering, GIS Dept.

K.N.Toosi University of Technology

Sejong University, South Korea

Tel | (+9821) 8887-7070

Fax | (+9821) 8878-6213

Mobile (Korea) |+8210 4253 5313

Official Email    | [email protected] 
Personal Email | [email protected]
Web | www.ubgi.ir

################################################

Reviewer 1

The Statement:

“The results are clearly presented.

Response:

We appreciate the reviewer for his/her valuable time for reviewing our manuscript. It is our pleasure and thanks to the reviewer that was satisfied with respect to the structure of our manuscript. As the reviewer’ comment shows that the results of the research are acceptable and convincing. We would like to thank the reviewer for this positive comment on our manuscript.

Reviewer 2 Report

Authors present an interesting paper, related to an emergent topic. 

To someone with less knowledge on ontologies, maybe a bigger clarification on its benefits could be given, other than the fact that it can store the user´s preferences, as that can be done in many other ways.

Congratulations on the authors, for the clear writing and document structure.

My main issue may go with the focus on this particular journal.

Author Response

Dear Prof. Dr. Mostafa Bassiouni,

Editor-in-Chief,

Electronics ,

May, 2019

Re: (Manuscript ID: electronics-512692) A Context-Aware Ontology-Based Route Finding Algorithm for Self-Driving Tourists

We thank both reviewers for providing us highly constructive and insightful comments to improve our manuscript. We have carefully revised the manuscript, following the reviewers’ suggestions, and have responded in detail to each comment. The next section contains our point-by-point responses (in blue) and changes and referencing to the manuscript (in green), based on the reviewers’ comments (in italic). We believe that our manuscript has substantially improved and is more readable for broader audiences. We would like to thank you, Prof. Dr. Mostafa Bassiouni (Editor-in-Chief), related Section Editor-in-Chief and Ms. Coleen.Shi, M.S. (Assistant Editor) for following up this manuscript. We look forward to hearing from you. Also, we would be glad to respond to any further questions and comments that you may have.

Best Regards,

Abolghasem Sadeghi-Niaraki,

################################################

Dr. Eng. Abolghasem Sadeghi-Niaraki

Assistant Professor

Faculty of Geodesy and Geomatics Engineering, GIS Dept.

K.N.Toosi University of Technology

Sejong University, South Korea

Tel | (+9821) 8887-7070

Fax | (+9821) 8878-6213

Mobile (Korea) |+8210 4253 5313

Official Email    | [email protected] 
Personal Email | [email protected]
Web | www.ubgi.ir

################################################

Reviewer 2

The Statement:

“Authors present an interesting paper, related to an emergent topic.

Congratulations on the authors, for the clear writing and document structure.

My main issue may go with the focus on this particular journal.

Response:

We appreciate the reviewer for his/her valuable time for reviewing our manuscript and providing us useful comments; they have certainly improved our article.

The Comment:

To someone with less knowledge on ontologies, maybe a bigger clarification on its benefits could be given, other than the fact that it can store the user´s preferences, as that can be done in many other ways.

Response:

We thanks for the reviewer’s comment. We have applied this and have added additional information to the revised version of our manuscript. It reads (Page 4, Paragraph 3):  

To propose route finding algorithm for self-driving tourists, the ontology model are used to find route real-time based on user needs in every condition. As route finding algorithms usually take long time to find optimal route between two points due to the large amount of data (Shen and Ban, 2016), ontology facilitates the timeliness of services and provides the capability to make use of earlier information without re-processing. Moreover, the data in ontologies are stored in plain text formats such as OWL. Obviously, plain text formats require much lower storage space. Therefore, the problem regarding the high volume of data will be resolved.

Shen, J., & Ban, Y. (2016). Route Choice of the Shortest Travel Time Based on Floating Car Data. Journal of Sensors, 2016.

Reviewer 3 Report

Though the problem considered is interesting, the proposed approach seems
inappropriate for the following major reasons:
- lack of a clear definition of the involved objective(s) and constraints;
- lack of a justified solution approach: the authors mention Dijkstra's
algorithm, but if several points are to be visited another framework is needed
- lack of a convincing comparison to other existing approaches/tools.
In total, what is the added value of this manuscript over the body of literature
reviewed, for example, in this paper: https://doi.org/10.1016/j.ejor.2016.04.059
?
Replacing a well-defined notion of 'score' by a much-less-well-defined notion of
'ontology' does not create anything new.

Some other comments:

- p.7 Eqs (1),(2) seem quite arbitrary; what is the logic behind them? Doesn't a
high frequency of using some road means that it should be avoided (to avoid
getting stuck in a traffic jam)? Overall, it seems that the proposed approach
would "prefer" short heavily used roads, which translates to reality as
preferring small overcrowded passages over main roads. This phenomenon is also seen in Figures 8 and 9. Regarding these pictures and the comparison of the authors' results to the results from Google Maps, a valid conclusion can be made only if exactly the same traffic conditions are input to both approaches and many enough experiments are performed (traffic conditions are stochastic, after all).

- p.8 the procedure for searching the nearest points to source/destination in
the network is very inefficient. Also, it is sensitive to the absolute
orientation of the points in question. What influences the choice of the value
of h?

Author Response

Dear Prof. Dr. Mostafa Bassiouni,

Editor-in-Chief,

Electronics ,

May, 2019

Re: (Manuscript ID: electronics-512692) A Context-Aware Ontology-Based Route Finding Algorithm for Self-Driving Tourists

We thank both reviewers for providing us highly constructive and insightful comments to improve our manuscript. We have carefully revised the manuscript, following the reviewers’ suggestions, and have responded in detail to each comment. The next section contains our point-by-point responses (in blue) and changes and referencing to the manuscript (in green), based on the reviewers’ comments (in italic). We believe that our manuscript has substantially improved and is more readable for broader audiences. We would like to thank you, Prof. Dr. Mostafa Bassiouni (Editor-in-Chief), related Section Editor-in-Chief and Ms. Coleen.Shi, M.S. (Assistant Editor) for following up this manuscript. We look forward to hearing from you. Also, we would be glad to respond to any further questions and comments that you may have.

Best Regards,

Abolghasem Sadeghi-Niaraki,

################################################

Dr. Eng. Abolghasem Sadeghi-Niaraki

Assistant Professor

Faculty of Geodesy and Geomatics Engineering, GIS Dept.

K.N.Toosi University of Technology

Sejong University, South Korea

Tel | (+9821) 8887-7070

Fax | (+9821) 8878-6213

Mobile (Korea) |+8210 4253 5313

Official Email    | [email protected] 
Personal Email | [email protected]
Web | www.ubgi.ir

################################################

Reviewer 3

The Statement:

“Though the problem considered is interesting.

Response:

We appreciate the reviewer for his/her valuable time for reviewing our manuscript and providing us useful comments; they have certainly improved our article.

The proposed approach seems inappropriate for the following major reasons:

The Comment:

·         Lack of a clear definition of the involved objective(s) and constraints;

Response:

Thanks for pointing out this comment. The following information has been applied in the revised version of the manuscript. It reads (Page 3, Paragraph 5):

This paper aims to propose a context-aware route finding algorithm based on ontology models of driver’s experiences for self-driving tourists. This algorithm considers two kinds of trips including business and touristic trips. In the proposed algorithm, if the location and destination are in the ontology of drivers’ experiences, the optimal route between location and destination will be presented for the user by retrieving route segments from ontology without any processing and if the location or destination doesn’t exist in the drivers’ experiences ontology, a part of the route will be calculated based on ontology and another part based on Dijkstra algorithm. In this regards, two ontologies are designed: the ontology of drivers’ experiences and the ontology of required services for tourists. After designing both ontologies, the data are collected for each of them. The data needed for the ontology of drivers’ experiences are collected by an application. The data needed for the services ontology are collected by OSM and located in the related class. After collecting data, a route finding algorithm is designed according to both ontologies. Finally, the algorithm is implemented for Tehran, Iran and compared in two modes (peak and off-peak) using Google Maps in terms of travel time and travel length.

The Comment:

·         Lack of a justified solution approach: the authors mention Dijkstra's algorithm, but if several points are to be visited another framework is needed

Response:

The authors agree with the reviewer’s comment that Dijkstra's is not suitable for visiting several points. However, the proposed algorithm is not based on Dijkstra's algorithm. The main idea of the algorithm is retrieving the optimal route between every location and destination form the drivers’ experiences ontology. Only a part of the route will be calculated using Dijkstra's algorithm if the location or destination doesn’t exist in the drivers’ experiences ontology. It should be noted while Dijkstra's algorithm can be used for finding all pair shortest paths by running it for every pair point, it would be worst from time complexity (Demetrescu et al., 2006). Our proposed algorithm doesn’t this problem because it doesn’t need to run Dijkstra's algorithm for every point. In addition, as the user can visit several points in the touristic trip, additional information has been added based on the reviewer’ comment to the revised version of the manuscript. It reads (Page 10, Paragraph 3):

“In the touristic trip, the ontology-based route finding problem is based on the concept of divide and conquer approach (Mei at al., 2013). With this approach, the problem of route finding is divided into several small scale sub-problems and then solve each sub-problem respectively. After that, the results of sub-problems connect together to create the overall route.”

Mei, Y., Li, X., & Yao, X. (2013). Cooperative coevolution with route distance grouping for large-scale capacitated arc routing problems. IEEE Transactions on Evolutionary Computation, 18(3), 435-449.

Demetrescu, C., & Italiano, G. F. (2006). Experimental analysis of dynamic all pairs shortest path algorithms. ACM Transactions on Algorithms (TALG), 2(4), 578-601.

The Comment:

·         Lack of a convincing comparison to other existing approaches/tools. In total, what is the added value of this manuscript over the body of literature reviewed, for example, in this paper: https://doi.org/10.1016/j.ejor.2016.04.059 ?

Replacing a well-defined notion of 'score' by a much-less-well-defined notion of 'ontology' does not create anything new.

Response:                           

We thank the reviewer for these provided references. In this paper, references are related to optimization algorithms. Optimization algorithms represent an approximation of the result but route-finding algorithms such as Dijkstra’s algorithm present an exact result for the user. On the other hand, by using the optimization algorithms, there is no guarantee that the result is the exact result and the travel time is minimized as much as possible.

We have present an exact result for the user due to using the route-finding algorithm to minimize the travel time as much as possible. The main idea of our proposed algorithm is using a driver’s route experiences ontology in route finding. Ontology model is not only a conceptual model. Ontologies provide the capability to make use of earlier information without re-processing. This facilitates the timeliness of services. Moreover, the data in ontologies are stored in plain text formats such as OWL. Obviously, plain text formats require much lower storage space. Therefore, the problem regarding the high volume of data will be resolved.

Some other comments:

The Comment:

p.7 Eqs (1),(2) seem quite arbitrary; what is the logic behind them? Doesn't a high frequency of using some road means that it should be avoided (to avoid  getting stuck in a traffic jam)?Overall, it seems that the proposed approach would "prefer" short heavily used roads, which translates to reality as preferring small overcrowded passages over main roads. This phenomenon is also seen in Figures 8 and 9. Regarding these pictures and the comparison of the authors' results to the results from Google Maps, a valid conclusion can be made only if exactly the same traffic conditions are input to both approaches and many enough experiments are performed (traffic conditions are stochastic, after all).

Response:                           

We thank the reviewer for this insightful comment. We have used the frequency of each segment as its weight based on the research of Yuan et al. (2010). This reference has been added to the revised version of the manuscript. It reads (Page 7, Paragraph 2):

The edge’s weight is determined based on the idea of using the frequency of segments that are traveled by drivers. This idea was proposed by Yuan et al. (2010).

Regarding the equations, The Eq (1) is based on the Min-Max normalization method:

X’= (x-min(x))/(max(x)-min(x))

However, because for a segment which is not in the experimental way, f(ei) is zero, the numerator would be a negative number. In addition, in the Dijkstra algorithm, the weights cannot be negative values. Therefore, we have added 1 to both numerator and denominator to avoid negative results. It can be considered as the Eq(1):

Because in all cases f(ei)≥fmax, adding a constant to both numerator and denominator does not have any effect on the comparison and calculation of weights. As Dijkstra algorithm would rather a segment with lower weight than higher weight, we have used the Eq(2) to reverse the values.

   Eq. (2)

Based on the review’s comment, we have added some information for better clarification to the revised version of our manuscript. It reads (Page 7, Paragraph 3):

Where fn(ei) is the number of normal passages from ei, f(ei) is the number of passages from ei, fmin and fmax are the minimum and maximum number of passages from the edges. It should be noted, this formula is obtained based on Min-Max normalization method in which 1 value is added to both numerator and denominator to avoid negative results. For a segment which is not in the experimental way, f(ei) is zero. Therefore, fmin is zero. In general, the weight of a network graph is determined as follows to reverse the values:

The high frequency of using some roads by experienced drivers means that this road segment has lower traffic because experienced drivers use segments with lower traffic based on their previous experience. In figures 8 and 9, the majority of segments in the calculated route are in the main roads not secondary roads and just some parts are not in main roads which are related to the shortcuts which have been used by experienced drivers.

It reads (Page 12, Paragraph 1, Line 13):

As the comparisons between the proposed algorithm and Google Maps are done in the same departure times, they have the same traffic conditions.

 The Comment:

p.8 the procedure for searching the nearest points to source/destination in the network is very inefficient. Also, it is sensitive to the absolute orientation of the points in question. What influences the choice of the value of h?

Response:                           

We appreciate the reviewer’s comment. The idea of finding the nearest points to source and destination is based on this fact that, we can search a limited area of the main road network (which is too complicated) in a short time and it increases the efficiency of Dijkstra algorithm in terms of presenting an exact route for the user in the shortest execution time. The comment has applied and the explanations regarding the h value have added to the revised version of the manuscript. It reads (Page 9, Paragraph 1):

 “As shown in Fig. 6, the distance between the source (S) and destination (D) is first considered as diagonal (d1) of a rectangle known as R1. Then the h value is added to d1 to each side of the rectangle in order to create a large rectangle of R2. Actually, the h value is used to define searching rectangle dimensions (R2) to find the nearest point(s) to source or destination in the Experimental network. If the h value is large, the processing volume is increased due to the increasing number of segments, and if the value of h is small, the nearest point(s) to the source or destination may not be found. In this study, the value of h was considered 0.01 d1 which was determined based on trial and error.  Therefore, the diagonal (h) of the final rectangle is d2= d1 + 2 h. Finally, the larger rectangle is searched to find the nearest point(s) to the source or destination.” 

Reviewer 4 Report

The title of the paper distorts its understanding . In fact, the authors use ontology to recommend intermediate points of the path. The path itself is constructed using the previously known shortest-path search algorithms on a graph. It is worth further processing the title.

In Figure 1, it is not clear what the names of the concept classes mean. Intuitive, that this is the starting and ending point of the path, however, apparently, it is worth explicitly indicating this in the description text. Also, it is worth giving an example of individuals for the route class with all the properties of an individual, so that the example is more understandable.

It is not clear where the algorithm itself. Section 2 contains a description of possible cases, formulas for calculating the weights of the graph, but formally the algorithm is not presented. Authors are recommended to explicitly describe the algorithm in the form of pseudocode or flowchart.

The experiment did not show the criteria for choosing the path built on Google maps. Depending on the criterion - the shortest time or path, with or without traffic jams, Google provides different routes, each of which should be compared with that obtained using the proposed algorithm.

The article also requires thorough proofread. Page 1 line 40 - Petrina. In some places in text, OWL is represented as owl or Owl. As well as Api

Author Response

Dear Prof. Dr. Mostafa Bassiouni,

Editor-in-Chief,

Electronics ,

May, 2019

Re: (Manuscript ID: electronics-512692) A Context-Aware Ontology-Based Route Finding Algorithm for Self-Driving Tourists

We thank both reviewers for providing us highly constructive and insightful comments to improve our manuscript. We have carefully revised the manuscript, following the reviewers’ suggestions, and have responded in detail to each comment. The next section contains our point-by-point responses (in blue) and changes and referencing to the manuscript (in green), based on the reviewers’ comments (in italic). We believe that our manuscript has substantially improved and is more readable for broader audiences. We would like to thank you, Prof. Dr. Mostafa Bassiouni (Editor-in-Chief), related Section Editor-in-Chief and Ms. Coleen.Shi, M.S. (Assistant Editor) for following up this manuscript. We look forward to hearing from you. Also, we would be glad to respond to any further questions and comments that you may have.

Best Regards,

Abolghasem Sadeghi-Niaraki,

################################################

Dr. Eng. Abolghasem Sadeghi-Niaraki

Assistant Professor

Faculty of Geodesy and Geomatics Engineering, GIS Dept.

K.N.Toosi University of Technology

Sejong University, South Korea

Tel | (+9821) 8887-7070

Fax | (+9821) 8878-6213

Mobile (Korea) |+8210 4253 5313

Official Email    | [email protected] 
Personal Email | [email protected]
Web | www.ubgi.ir

################################################

Reviewer 4

We appreciate the reviewer for his/her valuable time for reviewing our manuscript and providing us useful comments; they have certainly improved our article.

The Comment:

The title of the paper distorts its understanding . In fact, the authors use ontology to recommend intermediate points of the path. The path itself is constructed using the previously known shortest-path search algorithms on a graph. It is worth further processing the title.

Response:                           

We thanks the reviewer’s comment. To make the title clear, it has been replaced with the following title in the revised version of the manuscript.

“A Context-Aware Route Finding Algorithm for Self-Driving Tourists using Ontology”

The Comment:

In Figure 1, it is not clear what the names of the concept classes mean. Intuitive, that this is the starting and ending point of the path, however, apparently, it is worth explicitly indicating this in the description text. Also, it is worth giving an example of individuals for the route class with all the properties of an individual, so that the example is more understandable.

Response:                           

Thank for the reviewer’ suggestions. We have improved the paragraph in the revised version of the manuscript by adding explanations of the ontology classes, properties, and individuals as following. It reads (Page 4, Paragraph 3):

“Fig. 1 shows the drivers’ experiences ontology. This ontology has two main classes of experimental and non-experimental routes. Each of these classes covers two subclasses of traffic and non-traffic. The experimental route class and its related subclasses are routes that drivers have traveled at peak traffic times (7-10 a.m. and 4-8 p.m.) and off-peak traffic times. Each drivers’ route is defined by a separate class. According to the departure time, it is also defined as a subclass of traffic or without traffic experimental routes classes. If the route is a subclass of without traffic class, the name of the class will be location_destination, and if it is a subclass of the traffic class, the class name will be location_destination_t in which t represents the traffic. The individuals of this class are route segments and the ID of each route segment is the individual’s name. X and Y coordinates are the individual’s data properties. For example, if the driver’ route between Poonak and Tajrish is considered at the traffic time, the class and OWL file named Poonak_Tajrish_t is created as a subclass of traffic class in drivers’ experiences ontology. In this class, each segment of the route, which should be traveled to reach Tajrish from Poonak at the peak traffic time, is defined as an individual. The individual of the first segment of Poonak_Tajrish_t class is as follows:

<owl:NamedIndividual   rdf:about="http://www.co-ode.org/ontologies/Poonak_Tajrish_t.owl#0">

<y_coor   rdf:datatype="http://www.w3.org/2001/XMLSchema#integer">3960467</y_coor>

<x_coor   rdf:datatype="http://www.w3.org/2001/XMLSchema#integer">535432</x_coor>

</owl:NamedIndividual>

Non-experimental routes include routes obtained through the route finding algorithm. According to the departure time, each new route, generated by the route finding algorithm, is defined as with traffic or without traffic subclasses of the non-experimental routes class. If the route is a subclass of without traffic class, the name of the class will be location_destination_n in which n represents non-experimental, and if it is a subclass of the traffic class, the class name will be location_destination_n_t. The individuals of each route are defined in a separate file as a class with a similar name to its route class in the drivers’ experience ontology. The reason for this work is preventing the ontology from complexity and increasing the search speed in the ontology file.”

The Comment:

It is not clear where the algorithm itself. Section 2 contains a description of possible cases, formulas for calculating the weights of the graph, but formally the algorithm is not presented. Authors are recommended to explicitly describe the algorithm in the form of pseudocode or flowchart.

Response:                           

We thanks for the reviewer’ comment. The algorithms have explained in the “2.4.1. Business trip” and “2.4.2 touristic trip” sections. To make them clear, the pseudocode of the business trip algorithm has been added to the revised version of the manuscript. It reads (Page 8 and Page 9):

Figure 5. Business trip algorithm

Figure 7. Touristic trip algorithm

Note: Please check the Fig. 5 and Fig. 7 in the attached track change file. 

The Comment:

The experiment did not show the criteria for choosing the path built on Google maps. Depending on the criterion - the shortest time or path, with or without traffic jams, Google provides different routes, each of which should be compared with that obtained using the proposed algorithm.

Response:                           

Thank you for pointing out this concern to us. The comment has been applied in the revised version of the manuscript. It reads (Page 12, Paragraph 1):

As the Google Maps navigation system uses traffic information, the routes which were calculated by Google Maps with traffic information are compared with the routes of the proposed algorithm in both traffic times and non-traffic times. In this regards, the fastest route of Google Maps is used to compare with the results of the algorithm based on time and distance.

We had limitations because we didn’t have other real data to compare the proposed algorithm based on different criteria. Based on the reviewer’s comment, we have added the future work to the revised version of the manuscript. It reads (Page 18, Paragraph 1):

In addition, it is best to compare the results of the proposed algorithm with the routes which are obtained based on different criteria such as with traffic or without traffic.  

The Comment:

The article also requires thorough proofread. Page 1 line 40 - Petrina. In some places in text, OWL is represented as owl or Owl. As well as Api

Response:                           

Thank you for pointing out these mistakes. The probable editorial mistakes have been corrected in the revised version of the manuscript. Besides, this manuscript has already sent to English Proofreading agency.

Round 2

Reviewer 3 Report

The authors have handled the minor comments properly, but the main issues still remain: (a) the added value of this paper over a body of literature that I mentioned in the previous revision; (b) the appropriateness of the proposed approach.

The authors make an attempt to respond to these issues, but from their response it appears that they have no slightest understanding of the topic.

In particular:

Optimization algorithms represent an approximation of the result but route-finding algorithms such as Dijkstra’s algorithm present an exact result for the user. On the other hand, by using the optimization algorithms, there is no guarantee that the result is the exact result and the travel time is minimized as much as possible.

The reality is exactly the opposite: optimisation algorithms do guarantee the optimality (unless they are heuristic). Dijkstra's algorithm is one of them, by the way. The divide-and-conquer algorithms in many cases are heuristics (there are few exceptions, though), so they do not guarantee optimality.

Moreover, the data in ontologies are stored in plain text formats such as OWL. Obviously, plain text formats require much lower storage space. Therefore, the problem regarding the high volume of data will be resolved.

From the example provided in the paper it appears that OWL is an XML-based format, meaning that it is actually VERY inefficient in terms of storage space. Obviously.

Author Response

Dear Prof. Dr. Mostafa Bassiouni,

Editor-in-Chief,

Electronics,

June, 2019

Re: 2nd Round- (Manuscript ID: electronics-512692) A Context-Aware Ontology-Based Route Finding Algorithm for Self-Driving Tourists

We would like to express our special thanks to the reviewer#3’ efforts for re-evaluating our manuscript and offering us extremely valuable comments for Round 2. Based on the reviewer#4’ comments, we have again revised the manuscript. The next section contains our point-by-point responses (in blue) and changes and referencing to the manuscript (in green), based on the reviewers’ comments (in italic). We believe that our manuscript has substantially improved and is more readable for broader audiences. We would like to thank you, Prof. Dr. Mostafa Bassiouni (Editor-in-Chief), related Section Editor-in-Chief and Ms. Coleen.Shi, M.S. (Assistant Editor) for following up this manuscript. We look forward to hearing from you. Also, we would be glad to respond to any further questions and comments that you may have.

Note: Please also check the revised manuscript of the Round 2(track change version) in the attached file.

Best Regards,

Abolghasem Sadeghi-Niaraki,

################################################

Dr. Eng. Abolghasem Sadeghi-Niaraki

Assistant Professor

Faculty of Geodesy and Geomatics Engineering, GIS Dept.

K.N.Toosi University of Technology

Sejong University, South Korea

Tel | (+9821) 8887-7070

Fax | (+9821) 8878-6213

Mobile (Korea) |+8210 4253 5313

Official Email    | [email protected] 
Personal Email | [email protected]
Web | www.ubgi.ir

################################################

Reviewer 3

The Statement:

The authors have handled the minor comments properly, but the main issues still remain: (a) the added value of this paper over a body of literature that I mentioned in the previous revision; (b) the appropriateness of the proposed approach.

The authors make an attempt to respond to these issues, but from their response it appears that they have no slightest understanding of the topic.

Response:

We appreciate the reviewer for his/her valuable time for reviewing our manuscript and providing us useful comments for the second round; they have certainly improved our article.

The Comment:

In particular:

Optimization algorithms represent an approximation of the result but route-finding algorithms such as Dijkstra’s algorithm present an exact result for the user. On the other hand, by using the optimization algorithms, there is no guarantee that the result is the exact result and the travel time is minimized as much as possible.

The reality is exactly the opposite: optimisation algorithms do guarantee the optimality (unless they are heuristic). Dijkstra's algorithm is one of them, by the way. The divide-and-conquer algorithms in many cases are heuristics (there are few exceptions, though), so they do not guarantee optimality.

Response:

We agree with the respectful reviewer. We think based on our previous explanation some misunderstandings were happened. In fact, the authors considered two kinds of terms “Optimal Route Algorithm” and “Optimization algorithm”. We considered “Optimal Route Algorithm” as an algorithm (such as Dijkstra’s algorithm) which presents an optimal exact result for the user. On the other hands, we defined “Optimization algorithm” like a heuristic based algorithm which is not always guarantee optimality. Following the reviewer’s comment, the information has been added to the revised version of the manuscript. It reads (Page 3, Paragraph 2):

In addition, other studies conducted that used heuristic optimization algorithms such as Genetics and PSO, Gunawan et al. (2016) reviewed some of them. These methods that can find a feasible path with a very high probability (Mawale et al., 2013) have not discussed in this study. This study uses the optimal route algorithm e.g. Dijkstra that exactly minimizes the sum of the edge weights to find an optimal path (Mawale et al., 2013; Feng et al., 2012) as a part of the proposed algorithm.

Mawale, M. V., & Gandole, Y. B. (2013). Analysis of Optimal Route algorithm under Constraint Conditions. International Journal of Computer Science and Information Technologies, 2614-2619.

Feng, G., Makki, K., Pissinou, N., & Douligeris, C. (2002). Heuristic and exact algorithms for QoS routing with multiple constraints. IEICE Transactions on Communications, 85(12), 2838-2850.

Gunawan, A., Lau, H. C., & Vansteenwegen, P. (2016). Orienteering problem: A survey of recent variants, solution approaches and applications. European Journal of Operational Research, 255(2), 315-332.

We also changed the “route optimization” to “optimal rout finding” in the revised version of the manuscript to make it clear.

The Comment:

Moreover, the data in ontologies are stored in plain text formats such as OWL. Obviously, plain text formats require much lower storage space. Therefore, the problem regarding the high volume of data will be resolved.

From the example provided in the paper it appears that OWL is an XML-based format, meaning that it is actually VERY inefficient in terms of storage space. Obviously.

Response:

We thanks for the reviewer’ comment. We mean that the OWL files are lower storage than SHP files. To remove the ambiguity of the sentence, we have improved the paragraph in the revised version of the manuscript as following. It reads (Page 4, Paragraph 3):

As route finding algorithms usually take long time to find optimal route between two points due to the large amount of data (Shen and Ban, 2016), modeling driver’ spatial experiences using ontology facilitates the timeliness of services and provides the capability to make use of earlier information without re-processing. Moreover, the advantage of using ontology to storing data in  OWL files is to enhance sharing, reusing and processing domain knowledge (Hasani et al., 2015; Sureephong et al., 2008).

Sureephong, P., Chakpitak, N., Ouzrout, Y., & Bouras, A. (2008). An ontology-based knowledge management system for industry clusters. In Global Design to Gain a Competitive Edge (pp. 333-342). Springer, London.

Hasani, S., Sadeghi-Niaraki, A., & Jelokhani-Niaraki, M. (2015). Spatial data integration using ontology-based approach. The International Archives of Photogrammetry, Remote Sensing and Spatial Information Sciences, 40(1), 293.

Reviewer 4 Report

The authors had done good work on processing reviewers suggestions. More formalization had been provided as well as explanations of unclear moments.

There are some mistypes on fig. 5 and 7:

"destiuation" -> "destination" (5 times)

"shortest past" -> "shortest path" (4 times only on fig 5.)

Author Response

Dear Prof. Dr. Mostafa Bassiouni,

Editor-in-Chief,

Electronics,

June, 2019

Re: 2nd Round- (Manuscript ID: electronics-512692) A Context-Aware Ontology-Based Route Finding Algorithm for Self-Driving Tourists

We would like to express our special thanks to the reviewer#3’ efforts for re-evaluating our manuscript and offering us extremely valuable comments for Round 2. Based on the reviewer#4’ comments, we have again revised the manuscript. The next section contains our point-by-point responses (in blue) and changes and referencing to the manuscript (in green), based on the reviewers’ comments (in italic). We believe that our manuscript has substantially improved and is more readable for broader audiences. We would like to thank you, Prof. Dr. Mostafa Bassiouni (Editor-in-Chief), related Section Editor-in-Chief and Ms. Coleen.Shi, M.S. (Assistant Editor) for following up this manuscript. We look forward to hearing from you. Also, we would be glad to respond to any further questions and comments that you may have.

Note: Please also check the revised manuscript of the Round 2(track change version) in the attached file.

Best Regards,

Abolghasem Sadeghi-Niaraki,

################################################

Dr. Eng. Abolghasem Sadeghi-Niaraki

Assistant Professor

Faculty of Geodesy and Geomatics Engineering, GIS Dept.

K.N.Toosi University of Technology

Sejong University, South Korea

Tel | (+9821) 8887-7070

Fax | (+9821) 8878-6213

Mobile (Korea) |+8210 4253 5313

Official Email    | [email protected] 
Personal Email | [email protected]
Web | www.ubgi.ir

################################################

Reviewer 4

The authors had done good work on processing reviewers suggestions. More formalization had been provided as well as explanations of unclear moments.

We appreciate the reviewer for his/her valuable time for reviewing our manuscript and providing us useful comments for the second round; they have certainly improved our article.

The Comment:

There are some mistypes on fig. 5 and 7:

"destiuation" -> "destination" (5 times)

"shortest past" -> "shortest path" (4 times only on fig 5.)

Response:

Thank you for pointing out these mistakes. These mistakes have been corrected in the revised version of the manuscript.

Note: It seems Figure 5 and Figure 7 cannot be shown here. So please refer to the attached Track Changed file.

Round 3

Reviewer 3 Report

The authors have handled the minor comments properly, but the main issues still remain: (a) the added value of this paper over a body of literature that I mentioned in the 1st revision; (b) the appropriateness of the proposed approach.